# Phage Resistance Modulates *Escherichia coli* B Response to Metal-Based Antimicrobials

**DOI:** 10.3390/antibiotics14090942

**Published:** 2025-09-18

**Authors:** Franklin C. Ezeanowai, Akamu J. Ewunkem, Ugonna C. Morikwe, Larisa C. Kiki, Lindsey W. McGee, Joseph L. Graves, Liesl K. Jeffers-Francis

**Affiliations:** 1Biology Department, North Carolina Agricultural and Technical State University, 1601 E Market Street, Greensboro, NC 27411, USA; cfezeanowai@aggies.ncat.edu (F.C.E.); ucmorikwe@aggies.ncat.edu (U.C.M.); klchila@aggies.ncat.edu (L.C.K.);; 2Department of Biological Sciences, Winston Salem State University, 601 S Martin Luther King Jr. Drive, Winston Salem, NC 27110, USA; ewunkemaj@wssu.edu; 3Department of Biology, Biochemistry, Public Health, Quality Science, Earlham College, 801 National Road West, Richmond, IN 47374, USA; mcgeeli@earlham.edu

**Keywords:** *Escherichia coli* B, T4 phage, Iron (III), heavy metals, antimicrobial resistance, experimental evolution, genomic adaptation

## Abstract

**Background/Objective**: The rise of multidrug-resistant bacteria underscores the urgent need for alternative antimicrobial strategies. Metal-based compounds and bacteriophage (phage) therapy have emerged as promising candidates, but the evolutionary trade-offs associated with these selective pressures and their combination remain poorly understood. This study aimed to investigate how prior exposure to T4 phage influences *Escherichia coli* B’s subsequent adaptation to iron (III) stress and to assess the resulting phenotypic and genomic signatures of dual resistance. **Method:** In this study, we performed experimental evolution using *Escherichia coli* B to investigate adaptive responses under four conditions: control (LB broth), T4 phage-only, iron (III) sulfate-only, and sequential phage followed by iron (III) exposure. Each treatment consisted of ten independently evolved populations (biological replicates), all derived from a common ancestral strain and passaged daily for 35 days. Phage resistance evolved rapidly, with complete resistance observed within 24 h of exposure. **Results:** In contrast, iron-selected populations evolved tolerance to high iron concentrations (1000–1750 mg/L) over time at a cost to resistance in other metals (gallium and iron (II) and antibiotics (tetracycline). Notably, prior phage exposure altered these outcomes: phage/iron-selected populations retained phage resistance and iron tolerance but showed diminished resistance to iron (II) and distinct antibiotic sensitivity profiles. Whole-genome sequencing revealed stressor-specific adaptations: large deletions in phage receptor-related genes (*waaA* and *waaG*) under phage pressure, and selective sweeps in iron-adapted populations affecting regulatory and membrane-associated genes (*qseB*, *basR*, *aroK*, *fieF*, *rseB*, and *cpxP*). **Conclusions:** These results demonstrate that the sequence of environmental stressors significantly shapes phenotypic and genetic resistance outcomes. Our findings highlight the importance of fitness epistasis and historical contingency in microbial adaptation, with implications for the design of evolution-informed combination therapies.

## 1. Introduction

Bacteriophages (phages), viruses that infect bacteria, are key ecological drivers of microbial diversity and evolution, playing central roles in nutrient cycling, host–pathogen dynamics, and microbial community structure [1,2]. Increasingly, phages are being explored for therapeutic purposes, particularly as alternatives or adjuncts to antibiotics in the face of rising antimicrobial resistance (AMR) [3,4]. However, phage therapy is complicated by the ability of bacteria to rapidly evolve resistance to phages, often through modifications to cell surface receptors, changes in membrane permeability, or activation of stress response systems [5,6,7]. 

While phage resistance mechanisms have been well-characterized, there is growing interest in how these adaptations interact with other environmental stressors, including heavy metals. Iron, in particular, represents a biologically essential yet potentially toxic element for bacteria at higher concentrations. At physiological levels, iron is required for enzymatic activity and energy metabolism, but in excess, it promotes oxidative stress through Fenton reactions and disrupts key metabolic processes [8,9]. Bacteria exposed to high levels of iron often undergo membrane remodeling, activate detoxification pathways, and alter gene expression profiles to mitigate damage [10,11]. These changes may overlap with mechanisms used to resist phage infection, raising questions about how bacteria adapt when faced with both stressors.

Recent studies have explored the concept of pleiotropy and cross-resistance in this context. Adaptations to one selective pressure, such as iron toxicity or phage predation, can influence susceptibility to antibiotics, disinfectants, and other metals [6,12,13,14,15]. In a recent investigation, Jeje et al. [7] exposed *Escherichia coli* K-12 to iron (III) sulfate before challenging populations with phage T7. The resulting dual-resistant populations showed evidence of shared mutational targets (*rpoB*, *rpoC*, and *waaC*) and pleiotropic trade-offs, suggesting that adaptation to one stressor can shape responses to another. However, the influence of the selection sequence on these interactions remains unclear. Specifically, it is not known whether prior adaptation to phages constrains or facilitates subsequent resistance to metal stress, or whether dual resistance arises through similar or distinct genetic pathways depending on exposure order.

This study aimed to determine how prior adaptation of *Escherichia coli* B to T4 phage influences subsequent resistance to iron (III) stress, while also assessing whether dual resistance alters cross-resistance patterns to other metals and antibiotics through distinct evolutionary pathways. This work contributes to a broader understanding of microbial adaptation under complex environmental stress and offers insights relevant to the rational design of combination therapies involving phages and metal-based antimicrobials.

## 2. Results

### 2.1. Experimental Evolution Design

Figure 1 summarizes the experimental evolution framework. Ten replicate populations of *E. coli* B were derived from a common ancestor and evolved under four different conditions for 35 days:Control: Serially passaged in LB broth;Phage-selected: Exposed to T4 phage in LB broth daily;Iron (III)-selected: Cultured in LB broth with 1500 mg/L iron (III) sulfate;Phage/iron (III)-selected: First evolved for phage resistance in LB with T4 phage, then transferred to LB broth with 1500 mg/L iron (III) sulfate. All populations were transferred daily at a 1:100 dilution. Populations were archived and analyzed via phenotypic assays and whole-genome sequencing after 35 days of serial transfer.

### 2.2. T4 Phage Resistance

To assess the evolution of phage resistance, all populations were exposed to T4 phage and evaluated using plaque assays after 35 days of serial transfer. Phage resistance evolved rapidly in the phage-selected and phage/iron (III)-selected populations, with both groups showing complete resistance to T4 infection. Plaque assays revealed no visible plaques in these two treatment groups, indicating effective inhibition of phage replication (Figure 2A,B). By contrast, iron (III)-selected, control, and ancestral populations exhibited varying degrees of susceptibility. Among these, the ancestral population was the most vulnerable, followed by the iron (III)-selected and control groups, which showed partial plaque formation and reduced lytic activity. These results indicate that T4 phage resistance evolved independently of iron selection, and that prior adaptation to phage stress (as in the phage/iron group) was sufficient to confer robust resistance, even under subsequent metal pressure. These findings confirm the rapid and repeatable evolution of phage resistance in *E. coli* B when exposed to strong phage selection and validate the design of the phage and phage/iron selection regimes.

### 2.3. Dual Resistance to Iron (III) Sulfate and T4 Phage

To evaluate how well each population tolerated simultaneous exposure to iron (III) sulfate and T4 phage, we performed a dual-resistance assay using 1250 mg/L iron (III) sulfate, a concentration previously shown to inhibit growth across all groups, and a standardized T4 phage dose. Resistance was quantified by colony-forming unit (CFU) counts following 24 h incubation under combined stress. The phage/iron (III)-selected populations exhibited the highest survival, producing threefold more CFUs than any other group (Figure 3). This result suggests that sequential selection, first for phage resistance, followed by iron adaptation, enabled the evolution of robust dual resistance. In contrast, populations selected with only one stressor, phage-selected or iron (III)-selected, showed only modest gains in survival compared to the control. The ancestral populations remained the most susceptible, showing negligible growth under dual stress, as shown in Table 1.

### 2.4. Iron (III) Sulfate Tolerance

The 24 h growth of all experimental populations was assessed in LB broth supplemented with increasing concentrations of iron (III) sulfate (0–1750 mg/L). At concentrations ranging from 0 to 250 mg/L, the ancestor and control populations showed higher optical density (OD_600_) compared to the iron (III)-selected and phage/iron (III)-selected groups (Figure 4a). At concentrations of 1000 mg/L and 1500 mg/L, the iron (III)-selected and phage/iron (III)-selected populations displayed higher OD_600_ values than the phage-selected, control, and ancestor populations (Figure 4b). Growth at 1750 mg/L iron (III) sulfate was confirmed by serial dilution plating. Only the iron (III)-selected and phage/iron (III)-selected populations produced colonies, while no growth was observed for the ancestor, control, or phage-selected groups (Table 2). No visible growth was detected in any population at 2500 mg/L. Bonferroni multiple comparisons indicated that all selected populations (phage-selected, iron (III)-selected, and phage/iron (III)-selected) and the control differed significantly from the ancestor (*p* < 0.001). The selected populations also differed significantly from the control (*p* < 0.001). The difference between the phage/iron (III)-selected and phage-selected populations was not statistically significant (*p* = 0.276), while a small but significant difference was observed between iron (III)-selected and phage/iron (III)-selected (*p* = 0.038) (Table 3).

### 2.5. Heavy Metal Resistance

Growth of all populations was measured after 24 h in LB broth supplemented with increasing concentrations of four heavy metals: copper (II) sulfate, silver nitrate, iron (II) sulfate, and gallium (III) nitrate (Figure 5a–d). In copper (II) sulfate (Figure 5a), all populations exhibited growth from 60 to 750 mg/L. No growth was detected in any group at 1000 mg/L. In silver nitrate (Figure 5b), the control and phage-selected populations grew at 1 mg/L. No growth was observed in any population at 2 mg/L. In iron (II) sulfate (Figure 5c), the ancestor, control, and phage-selected groups showed higher OD_600_ values across 60–750 mg/L. At higher concentrations (1000–1750 mg/L), the iron (III)-selected and phage/iron (III)-selected populations showed increased OD_600_ values compared to the other groups. In gallium (III) nitrate (Figure 5d), the ancestor, control, and phage-selected populations exhibited greater OD_600_ values at concentrations ranging from 60 to 500 mg/L. At higher concentrations (750–2500 mg/L), the iron (III)-selected and phage/iron (III)-selected populations displayed higher OD_600_ values relative to the non-iron-selected groups. Growth was maintained in these two groups at concentrations up to 2500 mg/L, whereas OD_600_ values declined in the ancestor, control, and phage-selected populations at the same concentrations.

### 2.6. Antibiotic Resistance

Antibiotic resistance was evaluated by measuring 24 h growth (OD_600_) of all experimental populations in LB broth supplemented with increasing concentrations of four antibiotics: sulfanilamide, ampicillin, tetracycline, and chloramphenicol (Figure 6a–d). Sulfanilamide (Figure 6a). The ancestor and control populations showed higher OD_600_ values than all selected populations (phage-selected, iron (III)-selected, and phage/iron (III)-selected) at every tested concentration (60–2500 mg/L). Growth steadily declined in all groups as sulfanilamide concentration increased. At concentrations ≥ 1500 mg/L, iron (III)-selected and phage/iron (III)-selected populations exhibited the lowest OD_600_ values among all groups. Ampicillin (Figure 6b). The control and phage-selected populations maintained detectable growth at 1–5 mg/L. The ancestor, iron (III)-selected, and phage/iron (III)-selected populations showed minimal or no growth across the same concentration range. Growth in the control and phage-selected groups decreased gradually with increasing concentration, while growth in the iron-adapted groups was nearly absent at all tested concentrations. Tetracycline (Figure 6c). The phage-selected populations were capable of growth in concentrations from 1 to 6 mg/L. The phage/iron (III)-selected group grew at 1–2 mg/L but showed no growth at higher concentrations. The ancestor, control, and iron (III)-selected populations showed minimal or no growth at any tetracycline concentration tested (1–8 mg/L). Chloramphenicol (Figure 6d).

None of the populations exhibited growth at any of the tested concentrations (1–25 mg/L). OD_600_ values remained near baseline across all groups and concentrations.

### 2.7. Statistical Analysis of Growth Responses to Metals and Antibiotics

Two-way ANOVA was used to assess the effects of population, concentration, and their interaction on bacterial growth (OD_600_) across all tested heavy metals and antibiotics (Appendix A). For silver nitrate, a significant effect of population was observed (*F* = 76.874, *p* < 0.001), along with a significant interaction between population and concentration (*F* = 10.338, *p* = 0.002). However, no significant difference was found between the control and phage-selected populations (*F* = 1.802, *p* = 0.182). In response to sulfanilamide, the ancestor population demonstrated significantly higher tolerance compared to the control (*F* = 23.848, *p* < 0.001) and phage-selected populations (*p* < 0.001). Both concentration (*F* = 20.617, *p* < 0.001) and the population × concentration interaction (*F* = 2.231, *p* = 0.019) were significant, indicating concentration-dependent differences among lineages. For tetracycline, both the iron (III)-selected (*F* = 6.569, *p* = 0.011) and phage/iron (III)-selected (*F* = 22.847, *p* < 0.001) populations had significantly higher growth compared to the control. A strong interaction between population and concentration was also observed for tetracycline (*F* = 11.678, *p* < 0.001). For all remaining stressors tested—iron (II), iron (III), copper (II), gallium (III), ampicillin, and chloramphenicol—the main effect of population was statistically significant (*p* < 0.001), indicating differential growth responses among the experimental groups.

Bonferroni multiple comparisons for heavy metal exposure (Appendix A) showed that iron (III)-selected populations outperformed phage-selected populations in iron (II) sulfate. In gallium nitrate, the control population exhibited higher growth at lower concentrations, while phage/iron (III)-selected populations showed higher OD_600_ values at higher concentrations. For copper (II) sulfate, the iron (III)-selected group had significantly greater growth than the other groups. In silver nitrate, the phage-selected population outperformed the iron-adapted groups, while no significant difference was detected between the control and phage-selected populations. In antibiotic resistance testing (Appendix A), the ancestor population consistently showed higher growth in sulfanilamide compared to all evolved groups. The phage-selected population exhibited higher OD_600_ values in ampicillin and tetracycline than the iron (III)-selected and phage/iron (III)-selected populations. No statistically significant differences in chloramphenicol resistance were observed among any of the groups.

### 2.8. Genomic Analysis

Whole-genome sequencing was conducted for all experimental populations after 35 days of selection. Sequencing coverage ranged from 70× to 190× across replicates. Mutations were identified using the breseq 0.38.3 pipeline, and results are summarized in Figure 7 and Figure 8 and detailed in Appendix A.

In the ancestor population, consistent selective sweeps were detected compared to the reference genome in the 23S rRNA gene region (positions 4,091,103–4,407,064) and in the intergenic region between *mqo* and *eco*. Nine of ten ancestor replicates also exhibited a sweep in *pmrB*. One replicate (ANC8) displayed additional sweeps in *cynR* and *dcuS* (see Figure 7a). These mutations were considered part of the ancestral genetic background and were excluded from analyses of adaptive responses in treatment groups unless their frequency changed significantly under selection.

In the control group, one replicate (Ctrl3) showed a selective sweep in *pgap_annot_001232*, a gene encoding a terminase ATPase subunit family protein, which had also appeared as a polymorphism in the ancestor (see Figure 7b). Significant polymorphisms in the *rhsC* gene and in the intergenic region between *mqo* and *eco* were present in almost all control replicates (see Figure 8). The iron (III)-selected populations exhibited multiple selective sweeps. Selective sweeps were observed in *ydbD*, *qseB*, and *qseC*, as well as in the intergenic region between *fieF* and *cpxP* (see Figure 7d). Additional polymorphisms were identified in *basR*, *aroK*, *rseB*, *qseB,* and *dctA* (see Figure 8). The *rhsC* polymorphism was present in all iron (III)-selected replicates. A high-frequency polymorphism in *aroK* was observed in seven of nine replicates. One replicate (Fe4) displayed a polymorphism in *dctA*, and three replicates had polymorphisms in *rseB* (see Figure 8d).

In the phage-selected populations, selective sweeps were found in several loci associated with lipopolysaccharide biosynthesis. PS5 and PS9 harbored mutations in *waaA*, with PS5 showing a 3 bp deletion and PS9 a single nucleotide polymorphism (SNP). PS9 also contained two large deletions affecting multiple genes (Appendix A). Additional polymorphisms were observed in *rhsC*, the intergenic region between *mqo* and *eco*, and in ribosomal RNA loci. PS8 showed a high-frequency SNP in *asmA*, and PS1 displayed a SNP in *arcA*. PS7 carried a 1 bp deletion in *waaG* and a SNP in *aegA*. PS9 exhibited three high-frequency SNPs in *tRNA-Aspn* (see Figure 8c).

The phage/iron (III)-selected populations displayed shared and unique genomic features. All replicates carried two shared polymorphisms: a SNP in *rhsC* and an intergenic mutation between *mqo* and *eco*. Several iron-associated variants observed as polymorphisms in the iron (III)-selected group—such as in *basR*, *aroK*, and *rseB*—swept to fixation in multiple phage/iron (III) populations (see Figure 7e). New SNPs emerged in *rpoC* (in p/f2, p/f7, p/f10), *infC* (p/f4–p/f6), and *menH* (p/f4–p/f6) (see Figure 8e). A de novo SNP in *rseB* reached fixation in p/f8 and was present at high frequency in p/f1 and p/f2. No mutations that had swept in the phage-selected populations were detected in the phage/iron (III)-selected group. A complete list of selective sweeps, significant polymorphisms, affected genes, genomic positions, and functional annotations is provided in Appendix A. Genomic composition details for *E. coli* B ATCC 11303 are summarized in Appendix A.

## 3. Discussion

### 3.1. Phage Resistance and Sequence of Selection

Phage resistance evolved rapidly in *E. coli* B populations exposed to T4 phage, with no visible plaques observed in the phage-selected and phage/iron (III)-selected groups within 24 h. This finding is consistent with previous work demonstrating the speed at which phage resistance can emerge under strong selection [3,4]. Rapid resistance is often achieved through modification or loss of phage receptors, particularly components of the outer membrane such as lipopolysaccharides (LPS), which serve as attachment sites for many lytic phages, including T4 [16,17,18].

The sequencing results support this mechanism: mutations and deletions in genes related to LPS biosynthesis, such as *waaA*, *waaG*, and *rfaQ*, were observed in several phage-selected populations. Notably, these alterations were absent from the phage/iron (III)-selected group, despite their phenotypic resistance. This indicates that while both groups achieved phage resistance, the underlying genetic pathways differed—suggesting an effect of the selection sequence on the evolutionary trajectory.

The design of this study—where phage resistance was evolved before iron exposure—contrasts with Jeje et al. [7], where populations were first selected for iron (III) resistance before being exposed to phage T7. In that study, dual-resistant populations shared phage-associated mutations with phage-only populations, suggesting a degree of evolutionary parallelism. In contrast, the current study found no overlap in phage-resistance mutations between the phage-selected and phage/iron (III)-selected groups, even though both were resistant to phage. This divergence strongly supports the influence of fitness epistasis, where prior mutations (in this case from phage resistance) constrain or redirect future adaptive responses to a second selective pressure [5,6,9,19]. Moreover, the complete absence of phage-associated deletions (e.g., in *waaA* or *waaG*) in the phage/iron (III)-selected genomes suggests that the iron environment may have either purged these variants or selected for alternative compensatory adaptations that preserved phage resistance. This aligns with the hypothesis that membrane remodeling under metal stress may influence or interfere with classical phage resistance mechanisms [6,20,21].

Together, these results highlight the importance of selection order in shaping the genetic and phenotypic outcomes of adaptation. Although both phage-only and phage/iron (III)-selected populations achieved resistance, they did so through distinct genetic routes. These findings have implications for phage therapy in environments where bacteria are already under metal or chemical stress, as prior adaptation could shift evolutionary trajectories and resistance mechanisms in unexpected ways.

### 3.2. Iron (III) Tolerance and Growth Trade-Offs

Long-term exposure to iron (III) sulfate led to enhanced tolerance in both the iron (III)-selected and phage/iron (III)-selected populations. These groups were the only ones capable of growing at 1750 mg/L Fe (III), and they also exhibited higher OD_600_ values than all other populations at concentrations ≥1000 mg/L. This pattern is consistent with earlier studies that demonstrated adaptive tolerance to excess iron in *E. coli* K-12 MG1655 [6,7,15], as well as the broader literature on bacterial metal resistance mechanisms [1,2,5].

At lower iron concentrations (0–250 mg/L), however, the ancestor and control populations outperformed the iron-adapted groups. This result suggests a trade-off, where adaptation to high-iron environments comes at the cost of reduced performance under iron-replete or basal conditions—a phenomenon observed in previous work on *E. coli* adaptation to iron (II) and gallium (III) stress [6,10]. Iron stress is known to induce oxidative damage, disrupt metabolic regulation, and interfere with energy production, particularly through Fenton chemistry and iron–sulfur cluster destabilization [22]. As such, iron-adapted populations often exhibit rewiring of metal homeostasis, efflux pathways, and membrane transport [21,23], which may be metabolically costly or suboptimal under standard growth conditions.

The differences between this study and Jeje et al. are also noteworthy. In that study, *E. coli* K-12 MG1655 adapted to iron (III) showed enhanced growth across a broader range of iron concentrations and did not exhibit reduced growth at low concentrations [7]. In contrast, *E. coli* B ATCC 11303 required higher concentrations to display a performance advantage. This may reflect differences in baseline iron content between media—Davis Minimal Broth in Jeje et al. (~5.5 mg/L of iron) vs. LB broth in this study (~10–15 mg/L of iron) [6,24]—as well as differences in strain-specific physiology, gene content, or regulatory architecture [25,26]. Interestingly, phage/iron (III)-selected populations performed similarly to iron (III)-selected populations at 1000 mg/L of iron (III), but outperformed them at 1500 mg/L. These results suggest that phage resistance does not preclude the evolution of high-level iron tolerance. However, the overlap in tolerance thresholds may also indicate a shared ceiling in adaptive potential under this selective regime.

Collectively, these findings confirm that *E. coli* B can evolve tolerance to toxic levels of iron (III) sulfate, but that this adaptation is conditionally beneficial and accompanied by performance costs in environments with low to moderate iron availability. These trade-offs may have implications for bacterial fitness in fluctuating environments and for the design of antimicrobial strategies involving metals.

### 3.3. Cross-Resistance and Pleiotropy in Metals and Antibiotics

Adaptation to iron (III) stress and phage pressure resulted in altered resistance patterns to other metals and antibiotics, revealing both synergistic and antagonistic pleiotropic effects. These outcomes highlight how selection for one stressor can inadvertently shape responses to unrelated environmental challenges. In metal cross-resistance assays, iron (III)-selected and phage/iron (III)-selected populations exhibited enhanced tolerance to iron (II) sulfate and gallium (III) nitrate at high concentrations (1000–2500 mg/L), consistent with prior reports of overlapping metal detoxification and uptake systems [2,5,22]. Gallium, in particular, mimics ferric iron (Fe^3+^) but cannot participate in redox cycling, and its toxicity is mediated by disruption of iron-dependent pathways. Resistance to gallium in *E. coli* has been linked to mutations in iron transport systems such as *fecA* and *feoB*, as well as general stress response regulators [6,10]. Despite these gains, iron-adapted populations displayed reduced growth in silver nitrate and at lower concentrations of gallium, a trade-off not observed in the control or phage-selected groups. Silver toxicity operates through different mechanisms, including membrane disruption and oxidative stress, and may conflict with adaptations optimized for iron stress [1,15]. Similar trade-offs were observed in previous studies, where gallium or silver resistance evolved independently of iron homeostasis and sometimes imposed metabolic burdens [5,27].

Antibiotic resistance profiles further underscored the complexity of these pleiotropic effects. In sulfanilamide assays, the ancestor and control populations showed the highest growth across all concentrations, while iron (III)-selected and phage/iron (III)-selected groups displayed reduced growth, particularly at higher concentrations. Sulfanilamide targets folate biosynthesis, and although it is not structurally related to metal ions, previous studies suggest that global regulatory changes or metabolic rerouting during metal stress may impair resistance to folate inhibitors [27,28]. In contrast, the phage-selected population exhibited greater resistance to ampicillin and tetracycline than the iron-adapted groups. This aligns with findings in Jeje et al. [7], where phage-selected populations of *E. coli* K-12 MG1655 showed elevated resistance to β-lactams and tetracyclines, possibly due to membrane remodeling or efflux activation during phage exposure [7,17]. However, none of the populations in this study showed growth in chloramphenicol, suggesting that resistance to this antibiotic may require more targeted adaptations not conferred by metal or phage selection.

Taken together, these results illustrate that pleiotropy can manifest differently depending on the stressor, genetic background, and sequence of exposure. While some adaptations led to broad-spectrum tolerance (e.g., gallium cross-resistance in iron-adapted populations), others revealed clear trade-offs (e.g., silver and sulfanilamide sensitivity). These findings reinforce that selection does not occur in isolation and that the consequences of adaptation must be evaluated across a wider ecological and pharmacological context [6,7,8,28,29].

### 3.4. Genomic Signatures of Adaptation

Whole-genome sequencing revealed distinct mutational patterns across treatment groups, reflecting the specific selective pressures encountered during the 35-day evolution experiment. Consistent with earlier studies on bacterial adaptation to environmental stress [5,6,15], mutations were concentrated in genes involved in metal homeostasis, membrane remodeling, and transcriptional regulation. In the iron (III)-selected populations, selective sweeps and high-frequency polymorphisms were consistently observed in several key loci. These included genes involved in two-component regulatory systems (*basR*, *qseB*, *qseC*), metal ion transport (*fieF*), and envelope stress responses (*cpxP*, *rseB*) [20,21,22,23,28]. The *basR* gene, part of the BasSR two-component system, modulates outer membrane lipid remodeling and antibiotic resistance, particularly under iron and antimicrobial peptide stress [10,30]. Polymorphisms in *qseB* and *qseC* have also been linked to iron sensing and virulence regulation in uropathogenic *E. coli* [20], and their repeated appearance across iron-selected replicates suggests a central role in coordinating adaptation to iron toxicity.

A high-frequency polymorphism in *aroK*, a shikimate kinase required for aromatic amino acid biosynthesis, was found in seven of nine iron-adapted replicates. Previous work has shown that *aroK* expression is sensitive to intracellular iron levels and may represent a metabolic adjustment under iron-rich or iron-limiting conditions [28]. Additional mutations were observed in *dctA*, a C4-dicarboxylate transporter linked to heme biosynthesis and iron flux [31], and in *rseB*, a regulator of the sigma E envelope stress response pathway [21]. The widespread fixation of these variants across independent replicates indicates strong parallel selection for regulatory remodeling and oxidative stress resistance.

In contrast, the phage-selected populations exhibited mutations primarily associated with surface structure and phage attachment. Selective sweeps and high-frequency polymorphisms were detected in lipopolysaccharide biosynthesis genes such as *waaA*, *waaG*, and *rfaQ*, as well as in outer membrane proteins (*asmA* and *aegA*) and regulatory genes like *arcA* [17,18]. PS5 and PS9 carried deletions affecting multiple LPS-associated genes, consistent with previous findings that loss-of-function mutations in *waa* loci confer resistance to T-even phages [17]. The presence of intergenic SNPs between *waaA* and *rfaQ*, and deletions in *waaG*, suggests modification of LPS core oligosaccharides, which serve as T4 phage receptors [18]. These results align with classical models of phage resistance through receptor inactivation [16,17] and reinforce the notion that membrane structure is a primary target of selection under phage attack.

The phage/iron (III)-selected populations showed a distinct mutational profile, with some overlap with iron-only selections (e.g., *basR*, *aroK*, and *rseB*) but no shared deletions or SNPs in phage receptor genes observed in the phage-only group. This absence of convergence suggests that prior phage resistance altered the genetic background such that subsequent iron adaptation favored different evolutionary solutions. Interestingly, *rpoC*, encoding the β′ subunit of RNA polymerase, appeared as a polymorphism in several phage/iron (III)-selected replicates but not in the phage-selected group. Mutations in RNA polymerase subunits are commonly associated with pleiotropic adaptation and transcriptional rewiring under stress [11,13,32], and may reflect downstream optimization following initial phage resistance.

Across all groups, polymorphisms in *rhsC*, a gene associated with efflux and general stress resistance [27], were widespread and often found at intermediate to high frequencies. This finding is consistent with previous studies identifying *rhs* family genes as targets during long-term adaptation to antibiotics and metal exposure [6,26]. Overall, the observed mutations reflect modular, stress-specific responses that vary according to the sequence and type of selection. These genomic data support a model in which adaptation to phage and metal stress follows largely non-overlapping paths. While both stressors ultimately influence envelope structure and regulatory architecture, the specific mutational routes differ depending on prior selection history. This underscores the role of historical contingency and fitness epistasis in shaping bacterial evolution under complex environmental conditions [6,9,19,33].

### 3.5. Fitness Epistasis and Evolutionary Trajectories

The comparative genomic and phenotypic analyses presented in this study highlight the significant role of fitness epistasis—the context-dependent effect of mutations—on bacterial adaptation to sequential environmental stressors. Despite facing the same final selection environment (Iron (III) sulfate), the phage/iron (III)-selected populations evolved along a different trajectory than the iron (III)-selected group, as evidenced by distinct mutational signatures, differential cross-resistance profiles, and unique fitness trade-offs. These outcomes reinforce the principle that historical contingency, or the order of selective pressures, can constrain or redirect evolutionary pathways [6,9,19]. In this study, phage resistance evolved first, followed by iron stress. The resulting phage/iron (III)-selected populations retained resistance to phage but did not exhibit the large deletions or receptor-modifying mutations seen in the phage-only group. Instead, they acquired mutations in iron-associated loci (*basR*, *aroK*, and *rseB*), suggesting that the prior phage-adapted genetic background shaped which subsequent mutations were beneficial or tolerated. This outcome contrasts with the results of Jeje et al. [6,7], where *E. coli* K-12 MG1655 populations were first adapted to iron (III) and then exposed to phage. In that study, dual-resistant populations displayed mutational overlap with both iron- and phage-only groups, including shared deletions in LPS biosynthesis genes. In the present study, no phage-associated mutations from the phage-only treatment were found in the phage/iron (III)-selected populations, despite phenotypic resistance to T4. This lack of convergence highlights how early mutations can preclude subsequent evolutionary paths that would otherwise be accessible—a hallmark of sign epistasis. Additionally, some mutations that initially swept to fixation in the iron (III)-selected populations (e.g., in *qseB*, *qseC*, *ydbD*) were lost or did not persist in the phage/iron (III)-selected group, indicating that these variants were either neutral or deleterious in the altered genetic context. In contrast, variants in *basR* and *aroK*, initially polymorphic or subdominant in iron-only treatments, swept to fixation in the phage/iron (III) background. These dynamics illustrate that adaptation is not only driven by environmental stressors, but also by the internal genomic landscape shaped by prior selection events [9,19,33].

From an applied perspective, these findings carry important implications for the design of combination therapies involving phages and metal-based antimicrobials. The results suggest that treatment sequence matters: the evolutionary outcome of phage–metal combinations is not simply additive but instead shaped by interaction effects between resistance mechanisms. These insights align with recent studies in experimental evolution and systems biology, which show that sequential therapies can promote or constrain multi-resistance depending on the order and intensity of stress exposure [6,7,29]. In summary, the evolutionary trajectories observed in this study underscore the importance of considering epistasis, pleiotropy, and selection history when predicting microbial resistance outcomes. Understanding these dynamics is critical for developing sustainable therapeutic strategies that minimize resistance evolution while maximizing treatment efficacy.

## 4. Materials and Methods

### 4.1. Media Conditions and Experimental Layout 

*Escherichia coli* B (ATCC 11303) and T4 phage strains were obtained from ATCC (Manassas, VA, USA). The genome of *E. coli* B (ATCC 11303) is composed of 4,622,284 base pairs, encoding 4494 genes. A total of ten *E. coli* B colonies were cultured in 10 different sterile 50 mL flasks containing 30 mL LB broth and were incubated at 37 °C at 150 rpm for 24 h (denoted as Ancestor). The Ancestor culture grown for 24 h (25 mL) was stored in 50% (*v*/*v*) glycerol at a 1:1 ratio and frozen at −80 °C. The reference genome (ATCC 11303) can be accessed at https://www.ncbi.nlm.nih.gov/datasets/genome/GCA_034424725.1/ (accessed on 12 September 2025).

#### Experimental Populations

The control group was set up by transferring 0.1 mL of each overnight Ancestor culture to 10 sterile 50 mL flasks containing 9.9 mL of LB broth. The phage-selected populations were evolved by inoculating 0.1 mL of T4-phage and 0.1 mL of each *E. coli* B population from the ancestor culture in 9.8 mL of LB broth. The iron (III)-selected populations were evolved by inoculating 0.1 mL of the overnight culture (Ancestor culture) into 9.9 mL of LB broth containing 1500 mg/L iron (III) sulfate. The phage-iron (III)-selected populations were evolved by inoculating 0.1 mL of overnight culture of the phage-selected populations (phage-selected culture) into 9.9 mL of LB broth containing 1500 mg/L iron (III) sulfate (see Figure 1). All the groups were incubated at 37 °C at 150 rpm for 24 h. These cultures were grown for 24 h, from OD_600_ ≈ 0.05 cells/mL at hour zero to OD_600_ ≈ 0.735 cells/mL at 24 h. The control and phage-selected cultures were propagated by daily transfers of 0.1 mL of each overnight culture into fresh 9.9 mL LB broth, while the iron (III)-selected and phage/iron (III)-selected groups were propagated by daily transfers 0.1 mL overnight culture of iron (III)-selected, and phage/iron (III)-selected into a fresh 9.9 mL LB broth containing 1500 mg/L of iron (III) sulfate for 35 days of regrowth before selection for iron (III) resistance began. The phage titer was determined (3.4 × 10^6^ pfu) and stored temporarily at −20 °C for short-term usage. For long-term storage, aliquots of filtered T4-phage were mixed with 50% (*v*/*v*) glycerol in a ratio of 50:50 and stored at −80 °C. The T4-phage stock was diluted using 0.86% (*w*/*v*) NaCl, also called phage juice.

### 4.2. Determination of Minimal Inhibitory Concentration (MIC) of Iron (III) Sulfate

A Minimum Inhibitory Concentration (MIC) assay per [7], used different concentrations of heavy metals and antibiotics to determine the effect of phage and iron (III) selection pressure. The minimum inhibitory assay (MIC) assessed the growth rates of *E. coli* B by estimating their optical density at 600 nm (OD600) under varying concentrations of heavy metals and antibiotics. MIC is defined as the lowest Concentration of an antimicrobial that inhibits visible growth by ~90% following 24 h of incubation at 37 °C [14]. These values were determined by broth dilutions of iron (III) sulfate in a 96-well microtiter plate format consisting of ten concentrations (0–2500 mg/L) in triplicate. The plates were incubated at 37 °C and agitated at 150 rpm in a MAXQ 400 incubator shaker (Thermofisher Scientific, Waltham, MA, USA). The OD600 was measured at zero h and after 24 h of incubation at 37 °C. The initial MIC was determined to be 1250 mg/L. A sub-MIC value (1000 mg/L) was chosen to initiate the selection experiment. This value allowed for the initial growth of cultures without causing their extinction. This concentration was later increased to 1500 mg/L after 7 days of exposure.

### 4.3. Assessing T4-Phage Resistance E. coli B

Phage-resistant *E. coli* B populations were evolved by inoculating 100 µL of each *E. coli* B overnight ancestor culture into a fresh 9.8 mL sterile LB broth containing 100 µL of T4 phage. The cultures were incubated in a shaker incubator at 37 °C for 24 h at 150 rpm. Plaque assay was carried out to determine the susceptibility of the ten T4-phage-resistant *E. coli* B populations to lytic phage attack. Luria Broth (LB)-agar plates (Fisher Scientific, Fair Lawn, NJ, USA) were used because they provided the necessary contrast to visualize the plaques formed. Phage resistance was confirmed by a plaque assay and by evaluating the average plaque-forming unit (see Figure 2A,B).

### 4.4. Dual Resistance Assay: T4-Phage and Excess Iron (III)

A dual-resistance assay was performed to assess the tolerance of the ancestor, control, phage-selected, iron (III)-selected, and phage/iron (III) populations to excess iron (III) sulfate and T4-phage lysis (see Figure 3). A preliminary dose-response test demonstrated that 1250 mg/L of iron (III) sulfate inhibited growth in all populations. This Concentration was adopted for the combined iron (III)-phage challenge. 9.9 mL of LB broth containing 1250 mg/L of iron (III) sulfate and T4-phage (3.4 × 10^6^ pfu) was dispensed in 50 sterile 25 mL flasks, and 100 μL of each selected population (ancestor, control, phage-selected, iron (III)-selected, and phage/iron (III)) was inoculated in the 25 mL flask in 10 replicates and incubated in a shaker incubator for 24 h. After incubation, each selected population was serially diluted and plated on an LB agar plate, and the colony count was taken at a 10^−3^ dilution factor.

### 4.5. Phenotypic Analysis

Resistance to metals and antibiotics was determined by measuring 24 h growth via optical density in increasing concentrations of heavy metals and antibiotics for each selected population (ancestor, control, phage-selected, iron (III)-selected, and phage/iron (III)) at 35 days. Iron (III) sulfate resistance was tested from 0 to 1500. Due to iron (III) sulfate’s thick brownish color at 1750 and 2500 mg/L, measuring optical density was not feasible. Selection treatments were grown overnight at these concentrations and plated via serial dilution. Bacterial growth was confirmed, and colony-forming units were determined. Iron (II) sulfate resistance was tested from 0–1750 mg/L; Ga (III) nitrate resistance was tested from 0–2500 mg/L; Ag nitrate resistance was tested from 0–2 mg/L. Ampicillin resistance was tested from 0–5 mg/L; sulfonamide from 0–2500 mg/L; tetracycline from 0–8 mg/L; and chloramphenicol from 0–25 mg/L. Bacterial growth in Luria Broth was assessed by measuring turbidity at 600 nm at 24 h, using a 98-well plate Synergic Mx spectrophotometer (Biotek, Henrico, VA, USA) using clear polyester 98-well plates.

### 4.6. Genomic Sequencing

The whole genomic DNA extraction of the ancestors, control, phage-selected, iron (III)-selected, and phage/iron (III) (10 replicates per Population) after 35 days of culture was carried out using the EZNA Bacterial DNA extraction kit (Omega Bio-tek^®^, Norcross, GA, USA) following the manufacturer’s instructions per our previous studies [5,6,7]. The DNA concentrations were normalized according to the Illumina library prep protocol using the QuantiFluor^®^ dsDNA system (Illumina, San Diego, CA, USA). Genomic libraries were prepared using the Illumina DNA Prep kit, and strict adherence was maintained to the Illumina library prep protocol. The samples were sequenced using the Illumina NextSeq 2000 sequencing platform. The depth of coverage of the sequencing runs ranged from 70 to 190×, with most exceeding 120× coverage, and the read length per sample is 2 × 50 bp. Genomic variants were called using the breseq 36.1 pipeline per our previous studies [15]. The breseq pipeline uses three types of evidence to predict mutations: read alignments, missing coverage, and new junctions. Any reads that indicate a difference between the sample and the reference genome that cannot be resolved to describe precise genetic changes are listed as ‘unassigned’. The algorithm computes frequency by the number of reads that contain the de novo mutation.

Nine replicates of iron (III)-resistant populations were sequenced, designated ‘Fe_1_, Fe_2_, …, and Fe_9_’. Ten replicates of phage-resistant populations were successfully sequenced, designated ‘PS_1_, PS_2_, … and PS_10_’. Ten replicates of Phage/iron (III) resistant populations were codenamed ‘p/f_2_, p/f_2_, … and p/f_10_’, and ten controls were sequenced and designated ‘Ctrl_1_, Ctrl_2_, and Ctrl_10_’. Ten replicates of ancestor populations were successfully sequenced, designated ‘ANC_1_, ANC_2_, …, and ANC_10_’.

### 4.7. Statistical Analysis

A general linear model, SPSS version 29, was used to test 24 h growth by optical density across concentrations for metals (Fe III, Fe II, Ga III, Cu II, Ag I), T4 phage, and antibiotics (ampicillin, chloramphenicol, sulfanilamide, and tetracycline) for all selection treatments and controls. Bonferroni’s multiple comparisons test assessed the significance of the mean difference in 24 h growth by all metals and antibiotics for all selection treatments. Plots were made using GraphPad Prism (10th version) and R-studio (2025.05.1+513) statistical tools. Genomic variants were called using the breseq 36.1 pipeline per our previous studies [15].

## 5. Conclusions

This study demonstrates that *Escherichia coli* B can evolve resistance to both T4 phage and excess iron (III) sulfate under prolonged, sequential selection. The order of exposure—phage selection followed by metal stress—resulted in the development of dual-resistant populations with distinct phenotypic and genomic profiles compared to single-stressor or control groups. Resistance to phage emerged rapidly, consistent with previous findings [3,4,7], while iron adaptation required sustained selection and was associated with genomic changes in regulatory, membrane, and metal transport genes [20,22,23,30,31]. Cross-resistance testing revealed both synergistic and antagonistic pleiotropic effects. Iron-adapted populations displayed increased tolerance to iron (II) and gallium (III) at high concentrations [5,6], but reduced growth in silver nitrate and sulfanilamide [1,8,27]. In contrast, phage-selected populations showed greater resistance to ampicillin and tetracycline [7,17], but did not gain metal tolerance. Whole-genome sequencing confirmed that different selective pressures and genetic backgrounds produced divergent adaptive paths, with phage/iron (III)-selected populations retaining iron-associated mutations but lacking key phage-resistance deletions. These findings highlight the importance of fitness epistasis and selection order in shaping microbial adaptation. The evolutionary trajectory was not solely dictated by the final environment, but by the sequence of prior stress exposures that constrained or redirected available mutational paths [6,9,19]. Together, this study provides experimental evidence that the order and combination of selective pressures can lead to distinct evolutionary outcomes, even when phenotypic resistance appears similar. These insights underscore the need for evolution-informed design of combination therapies, particularly those involving phages and metal-based agents. Anticipating how stressor sequence and genetic background influence resistance may be key to developing robust strategies for slowing or preventing the emergence of multidrug-resistant pathogens. 

While this study provides new insights into how sequential exposure to phage and iron (III) stress shapes microbial adaptation, several limitations should be acknowledged. First, although whole-genome sequencing identified distinct mutational signatures across treatment groups, no functional validation experiments were performed to directly confirm the causal role of these mutations in resistance. To address this, we plan to perform RNA sequencing (RNA-seq) on all populations in future work to determine the functional relevance of these mutations and assess transcriptional changes underlying resistance. Second, our study focused on a single bacterial strain (*E. coli* B) and one phage (T4), which may limit the generalizability of these results to other bacterial–phage systems or clinical isolates. Expanding this framework to additional strains, phages, and metal-based stressors would strengthen the broader applicability of the findings. Finally, although we tested for cross-resistance to several metals and antibiotics, the scope of our testing was not exhaustive. Investigating a wider range of antimicrobials and environmental conditions will be important to fully characterize pleiotropic trade-offs. Together, these limitations highlight the need for future research to extend and validate our findings, while underscoring the value of experimental evolution in revealing how stressor sequence and genetic background influence adaptive trajectories.

## Figures and Tables

**Figure 1 antibiotics-14-00942-f001:**
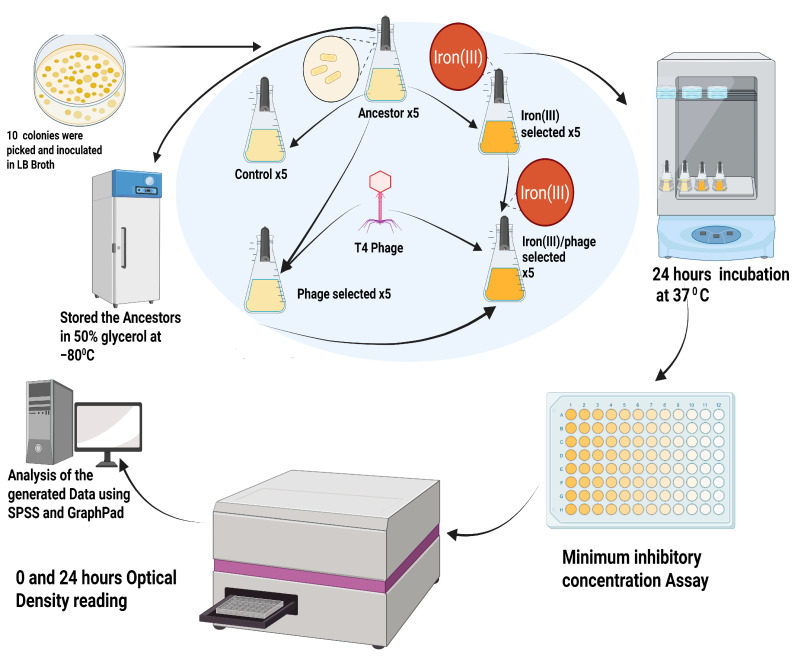
Experimental design for sequential selection. Schematic illustrating the 35-day experimental evolution of *Escherichia coli* B under four treatment conditions: Control (LB only), Phage-selected (T4 phage exposure), Iron (III)-selected (1500 mg/L Iron (III) sulfate), and Phage/Iron (III)-selected (sequential T4 phage exposure followed by Iron (III) sulfate). Created in BioRender. Ezeanowai, C. (2025) https://BioRender.com/mo69zd3 (accessed on 20 August 2025).

**Figure 2 antibiotics-14-00942-f002:**
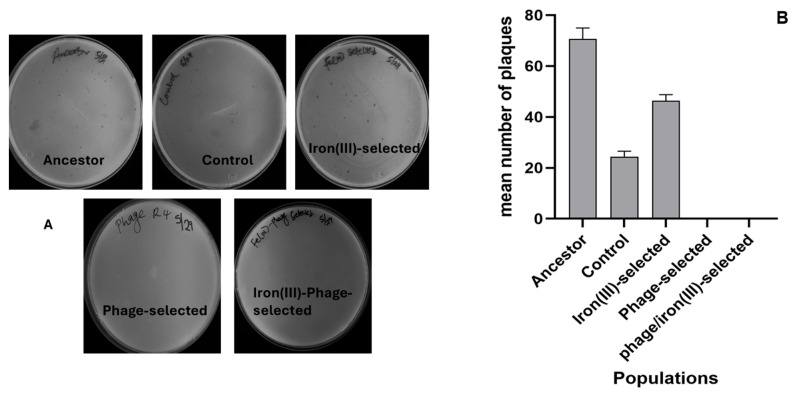
Qualitative and quantitative assessment of phage resistance. (**A**) Representative plaque assay plates following 24 h exposure to T4 phage across experimental populations. Ancestor and Control show clear plaques, iron (III)-selected shows partial resistance, and Phage-selected and Phage/iron (III)-selected complete resistance (no plaques). (**B**) Mean plaque counts per mL (±SEM) for each population. Phage-selected and Phage/iron (III)-selected populations showed no plaques; iron (III)-selected had intermediate resistance.

**Figure 3 antibiotics-14-00942-f003:**
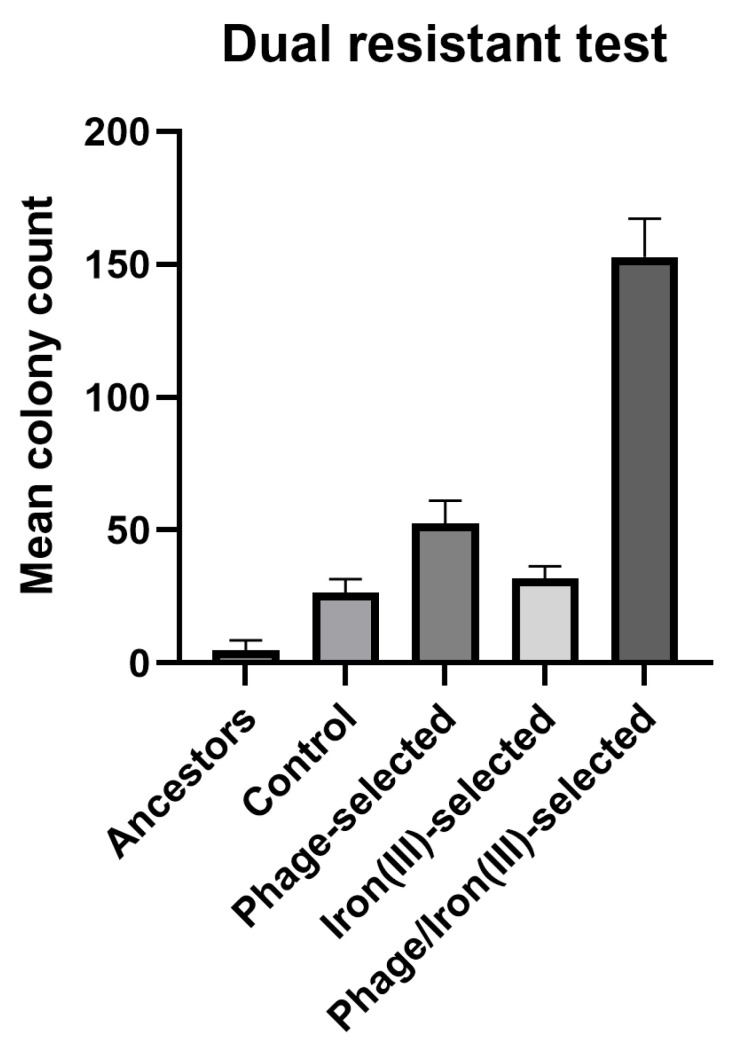
Dual resistance to T4 phage and iron (III) sulfate. Mean CFU/mL (±SEM) following 24 h co-exposure to 1250 mg/L iron (III) sulfate and T4 phage. The Phage/Iron (III)-selected population showed the highest survival; no growth was observed in the Ancestor and Control.

**Figure 4 antibiotics-14-00942-f004:**
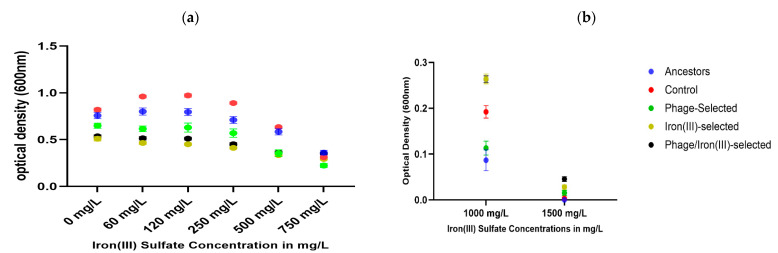
Iron (III) sulfate tolerance. (**a**) OD_600_ values after 24 h exposure to low concentrations of iron (III) sulfate (0–250 mg/L). Ancestor and Control populations performed best at these concentrations. (**b**) Growth at higher iron concentrations (1000–1500 mg/L) reveals enhanced tolerance in iron (III)-selected and Phage/iron (III)-selected populations.

**Figure 5 antibiotics-14-00942-f005:**
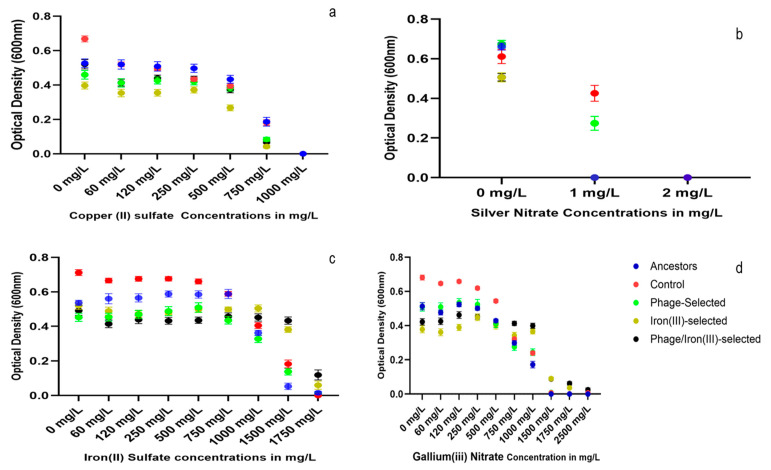
Cross-resistance test to heavy metals. OD_600_ values after 24 h exposure to: (**a**) Copper (II) sulfate (60–1000 mg/L), (**b**) Silver nitrate (1–2 mg/L), (**c**) Iron (II) sulfate (60–1750 mg/L), and (**d**) Gallium (III) nitrate (60–2500 mg/L). Iron-adapted populations showed enhanced tolerance to iron (II) and gallium at high concentrations but reduced tolerance to silver nitrate.

**Figure 6 antibiotics-14-00942-f006:**
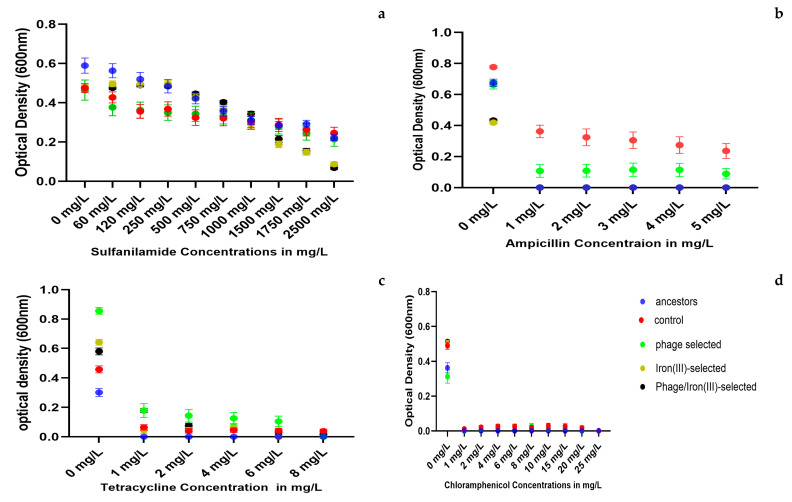
Antibiotic resistance profiles across populations. OD_600_ values after 24 h exposure to: (**a**) Sulfanilamide (60–2500 mg/L), (**b**) Ampicillin (1–5 mg/L), (**c**) Tetracycline (1–8 mg/L), and (**d**) Chloramphenicol (1–25 mg/L). Phage-selected populations showed higher resistance to ampicillin and tetracycline; iron-adapted groups were more sensitive to sulfanilamide.

**Figure 7 antibiotics-14-00942-f007:**
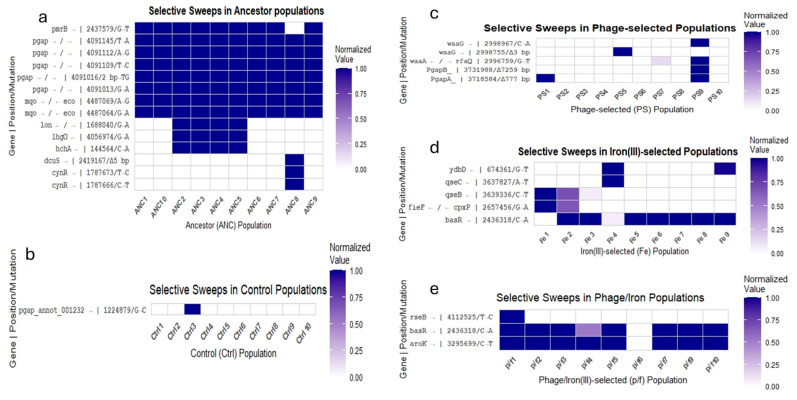
Heatmaps of selective sweeps. Genome-wide visualization of fixed mutations across replicate populations. (**a**–**e**) Correspond to Ancestor, Control, Phage-selected, iron (III)-selected, and Phage/iron (III)-selected groups. Color intensity reflects sweep frequency per gene or genomic region.

**Figure 8 antibiotics-14-00942-f008:**
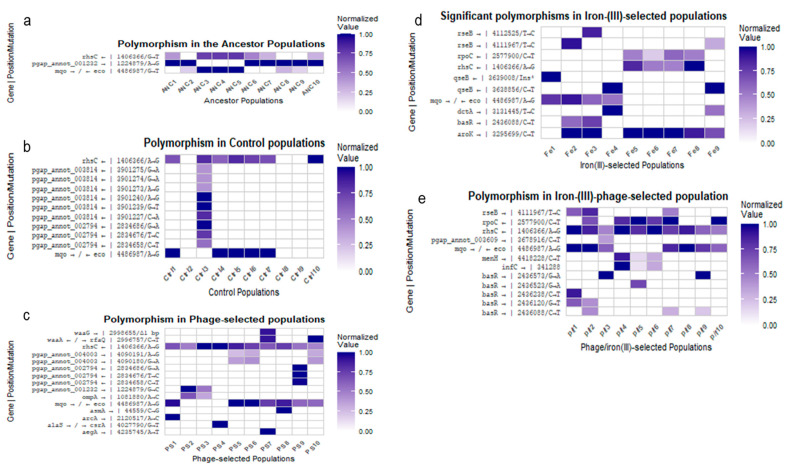
High-frequency polymorphisms (allele frequency ≥ 0.3). Heatmaps showing significant SNPs and indels across all treatment groups. (**a**–**e**) Represent Ancestor, Control, Phage-selected, iron (III)-selected, and Phage/iron (III)-selected populations. Mutated genes involved in membrane stress, metal detoxification, LPS biosynthesis, and transcriptional regulation are annotated.

**Table 1 antibiotics-14-00942-t001:** Relative fitness rankings of each population under T4 phage and iron (III) sulfate stress.Populations were ranked from 1 (highest resistance) to 5 (lowest resistance) based on their performance in plaque assays and CFU counts following exposure to 1250 mg/L iron (III) sulfate and T4 phage.

Population	Iron (III)	T4-Phage
Phage/iron (III)-selected	2	1
Phage	3	2
Iron (III)-selected	1	3
Control	5	4
Ancestor	4	5

**Table 2 antibiotics-14-00942-t002:** CFU/mL following overnight growth in 1750 mg/L iron (III) sulfate. Only iron (III)-selected and phage/iron (III)-selected populations demonstrated measurable growth at this concentration. Values represent mean colony counts (±SEM) from triplicate serial dilution plates. TMC—too many to count.

Dilution Factor	Ancestor (CFU/mL)	Control (CFU/mL)	Phage-Selected (CFU/mL)	Iron (III)-Selected (CFU/mL)	Phage/Iron (III)-Selected (CFU/mL)
10^−1^	0	0	0	TMC	1.77 × 10^4^ ± 3.51
10^−2^	0	0	0	6.53 × 10^5^ ± 15.37	2 × 10^4^ ± 1
10^−3^	0	0	0	5.67 × 10^5^ ± 1.53	0

**Table 3 antibiotics-14-00942-t003:** Bonferroni multiple comparisons of OD_600_ values in Iron (III) sulfate treatments. Post hoc pairwise comparisons among populations exposed to Iron (III) sulfate.

Fe(III) Sulfate Bonferroni’s Multiple Comparisons					95% Confidence Interval	
(I) Population	(J) Population	Mean Difference (I-J)	Std. Error	Sig.	Lower Bound	Upper Bound
Ancestor	Control	−0.0864 *	0.01145	0.000	−0.1186	−0.0542
	Phage	0.1154 *	0.01145	0.000	0.0832	0.1476
	Fe(III)-selected	0.1739 *	0.01145	0.000	0.1417	0.2061
	Phage/Iron (III)-selected	0.1406 *	0.01145	0.000	0.1084	0.1728
Control	Phage	0.2018 *	0.01145	0.000	0.1696	0.2340
	Fe(III)-selected	0.2603 *	0.01145	0.000	0.2281	0.2925
	Phage/Iron (III)-selected	0.2270 *	0.01145	0.000	0.1948	0.2592
Phage-selected	Fe(III)-selected	0.0585 *	0.01145	0.000	0.0263	0.0907
	Phage/Iron (III)-selected	0.0252	0.01145	0.276	−0.0069	0.0574
Fe(III)-selected	Phage/Iron (III)-selected	0.0332 *	0.01145	0.038	0.0010	0.0654

Based on observed means. The error term is Mean Square (Error) = 0.016. * The mean difference is significant at the 0.05 level.

## Data Availability

The SRA accession number for sequencing data from this study is PRJNA1321871 for iron (III)-selected, Phage-selected, /Phage/iron (III)-resistant, and control populations.

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
