# Peer review of "Phage Resistance Modulates Escherichia coli B Response to Metal-Based Antimicrobials"

_antibiotics, 2025, doi:10.3390/antibiotics14090942_

Round 1

Reviewer 1 Report

Comments and Suggestions for Authors

See attached review report

Author Response

Thank you very much for taking the time to review this manuscript. We have updated the references style in accordance with your manuscript guidelines. Please find the detailed responses below and the corresponding revisions/corrections highlighted (in yellow) in the re-submitted files.

Reviewer 1

TITLE:

  1. The authors used Escherichia coli, a Gram-negative bacterium, to study phage resistance, but extensively presented the conclusions. It would be helpful to consider whether the results might differ if a Gram-positive bacterium, such as Staphylococcus aureus, were used. If that is the case, the title could be revised to better reflect the actual scope of the study.

Comment: Thanks for your comment. We have revised the manuscript title to “STACKING THE STRESSORS: PHAGE RESISTANCE ALTERS ESCHERICHIA COLI B RESPONSE TO METAL-BASED ANTIMICROBIALS’ in lines 1-2

ABSTRACT:

  1. The problem statement and knowledge gap are clearly presented; however, the research objective is missing. The authors should consider adding the research objective at line 15.

Comment: We appreciate your observation regarding the absence of a clearly stated research objective. We have now added a sentence at the end of the "Background/Objective" section to explicitly state the aim of the study in lines 18-20.:
“This study aimed to investigate how prior exposure to T4 phage influences Escherichia coli B’s subsequent adaptation to Iron (III) stress and to assess the resulting phenotypic and genomic signatures of dual resistance.”

  1. Although the method used to address the objective is mentioned in line 15, it is too general. It should be refined to make it more concise and specific. Furthermore, the study does not indicate the number of samples used.

Comment: The methods described in the abstract have been revised to be more concise and specific, and the number of samples has been highlighted in lines 20-25:

“In this study, we performed experimental evolution using Escherichia coli B to investigate adaptive responses under four conditions: control (LB broth), T4 phage-only, iron (III) sulfate-only, and sequential phage followed by iron (III) exposure. Each treatment consisted of ten independently evolved populations (biological replicates), all derived from a common ancestral strain and passaged daily for 35 days.”

INTRODUCTION

  1. Lines 67–71 present the study objectives in an overly lengthy way and only mention iron, despite the investigation of other metal-based antimicrobials. The authors should combine the objectives into a single, clear statement that accurately includes all the antimicrobials studied.

Comment: Thank you for this helpful suggestion. We agree that the study objectives in lines 72–75 were presented in an overly lengthy way and placed too much emphasis on iron, without clearly capturing the broader scope of the work. We have revised this section to provide a single, concise objective statement that reflects both the primary selection stressors (T4 phage and iron(III)) and the inclusion of additional metals and antibiotics in cross-resistance testing.

The revised objective now reads as follows:
“This study aimed to determine how prior adaptation of Escherichia coli B to T4 phage influences subsequent resistance to iron(III) stress, while also assessing whether dual resistance alters cross-resistance patterns to other metals and antibiotics through distinct evolutionary pathways.”

RESULTS

  1. The authors should maintain consistency in referring to Iron(III) throughout the manuscript, as it currently appears in different formats (Iron(III) and Iron (III).

Comment: We thank the reviewer for pointing this out. The manuscript has been revised to ensure consistent use of “Iron (III)” throughout.

  1. Figure 2A (line 106) has poor resolution, making the plaques unclear to the reader.

Comment: Thank you for pointing this out. We have improved the brightness and contrast of Figure 2A to enhance clarity. The plaques are now more distinct and easily visible to the reader.

  1. It is unclear how resistance was evaluated, particularly how partial or intermediate, moderate resistance was determined, especially in the context of phage assessment. Measurements for resistance must be standardized.

Comment: Phage resistance was evaluated using plaque assays for all populations. If plaques were present, the population (ancestor, control, and iron(III)-selected) was considered susceptible to T4 phage lytic attack, whereas the absence of plaques indicated complete resistance (as in the phage-selected and phage/iron-selected groups). We did not measure partial or intermediate resistance; resistance was categorized strictly as either susceptible (plaques present) or resistant (no plaques observed). This has been clarified in Section 2.2 (T4 Phage Resistance).

  1. What is the basis of Figure 3 on dual resistance (line 129)? It appears to be based on the means of individual colony-forming units. How many replicates were performed, and why were statistical analyses not applied to support this conclusion?

Comment : The dual resistance assay was performed using 10 biological replicates per treatment group. Figure 3 presents the mean colony-forming units (CFUs) with standard error of the mean (SEM) to reflect variation across replicates.

  1. All tables should be revised to reduce lengthy narration, and all acronyms should be fully explained within the tables throughout the manuscript.

Comments: We appreciate the reviewer’s concern. All tables and figures in the manuscript were designed to succinctly explain the results. Acronyms are fully defined the first time they appear in each table or figure. We have carefully reviewed the manuscript again to ensure this consistency throughout.

  1. The artwork of Figures 5 and 6 can be improved by aligning the graphs and enlarging them to enhance visual clarity and improve the legibility of the legends.

Comments: We have revised Figures 5 and 6 by aligning the panels and enlarging the graphs to improve visual clarity and enhance the legibility of the legends.

  1. Under genomic analysis (line 242), it is unclear how many samples underwent whole-genome analysis.

Comments: Information on the number of replicates and sequencing methods is clearly described in Section 4.6 (Genomic Analysis). As stated, ten replicates were sequenced per treatment group, with sequencing depth ranging from 70× to 190×, with most exceeding 120× coverage. Figures 7 and 8 also display the number of replicates per population for transparency.

  1. The resolution of Figures 7 and 8 should be improved to enhance visual clarity and ensure all content is legible.

Comments: The resolution of Figures 7 and 8 has been improved to ensure that all labels and content are fully legible.

  1. The authors should provide the raw Optical Density (OD) values at 600 nm for heavy metal resistance (lines 163–178) in the supplementary materials.

Comments: This has been uploaded as supplementary.

Discussion

  1. The authors should acknowledge the limitations of their study, particularly in the genomic analyses in lines 403-457, as no experiments were conducted to validate the findings.

Comments: We agree with the reviewer’s suggestion and have revised the Discussion section (lines 636–650) to explicitly acknowledge the limitations of our genomic analysis and suggestions for future studies.

Materials and methods

  1. Were the sequences derived from whole-genome analysis deposited in a public repository? If so, specify the repository and provide the corresponding link or accession details.

Comments: The sequencing data have now been deposited in NCBI, and the SRA numbers (PRJNA132187) are available in the Data Availability section in lines 670-671.

Conclusion

  1. The conclusion reads more like an extension of the discussion rather than a clear summary of the study’s findings. The excessive citations obscure the key results. This section needs major revision and reorganization to improve clarity and readability.

Comments: The Conclusion section has been reorganized and revised to provide a clearer summary of the key findings, while reducing excessive citations. The revised version highlights the study's significant outcomes and their broader implications.

Reviewer 2 Report

Comments and Suggestions for Authors

The manuscript demonstrates the development of phenotypic and genetic metal-resistance  in E. coli B in the presence of T4 phage. The manuscript is engaging and well-written overall.

Strengths:

  1. The study is relevant, given the rise of antimicrobial resistance bacteria resulting in treatment failures and emergence of metal-based drugs and bacteriophages as alternative treatment strategies.
  2. Presenting a comprehensive summary of the entire research, the study meticulously details its background, methods, key results, and conclusions with clarity and conciseness.
  3. Based on whole-genome sequencing, the study presented comprehensive comparisons of the resistant bacterial populations at genetic level.

Areas for improvement:

  1. To enhance clarity and readability, it is recommended to standardize the font size and style across all figures
  2. Outlining the limitations of the study will help in detailing their potential impact on the findings and suggesting avenues for future research.

Author Response

Thank you very much for taking the time to review this manuscript. We have updated the references style in accordance with your manuscript guidelines. Please find the detailed responses below and the corresponding revisions/corrections highlighted (in yellow) in the re-submitted files.

Reviewer report 2

  1. To enhance clarity and readability, it is recommended to standardize the font size and style across all figures

Comment: We have updated the font size and style in accordance with the manuscript guidelines

  1. Outlining the limitations of the study will help in detailing their potential impact on the findings and suggesting avenues for future research.

Comment: We have updated the manuscript to include some limitations in our current study in lines 637-653. The revised text is as follows: “ While this study provides new insights into how sequential exposure to phage and iron (III) stress shapes microbial adaptation, several limitations should be acknowledged. First, although whole-genome sequencing identified distinct mutational signatures across treatment groups, no functional validation experiments were performed to directly confirm the causal role of these mutations in resistance. To address this, we plan to perform RNA sequencing (RNA-seq) on all populations in future work to determine the functional relevance of these mutations and assess transcriptional changes underlying resistance. Second, our study focused on a single bacterial strain (E. coli B) and one phage (T4), which may limit the generalizability of these results to other bacterial–phage systems or clinical isolates. Expanding this framework to additional strains, phages, and metal-based stressors would strengthen the broader applicability of the findings. Finally, although we tested cross-resistance to several metals and antibiotics, the scope was not exhaustive. Investigating a wider range of antimicrobials and environmental conditions will be important to fully characterize pleiotropic trade-offs. Together, these limitations highlight the need for future research to extend and validate our findings, while underscoring the value of experimental evolution in revealing how stressor sequence and genetic background influence adaptive trajectories.”

Reviewer 3 Report

Comments and Suggestions for Authors

Reviewer Comments

  • In the introduction section please briefly discuss some of the crucial genes involved in the study.
  • In the sequencing section please include the sequencing depth and also indicate if the reads were short or long with their size/size range.
  • Please indicate which is the reference genome chosen.
  • Please explain why only certain metals were used and substantiate the choice of their concentrations with suitable references.
  • Fig 1: Escherichia coli should be in italic.
  • See Figure 8e).. Delete the extra full stop.

Author Response

Thank you very much for taking the time to review this manuscript. We have updated the references style in accordance with your manuscript guidelines. Please find the detailed responses below and the corresponding revisions/corrections highlighted (in yellow) in the re-submitted files.

Reviewer report 3

1. In the introduction section, please briefly discuss some of the crucial genes involved in the study.

Comment: We have updated the introduction to include “The resulting dual-resistant populations showed evidence of shared mutational targets (rpoB, rpoC, and waaC)” from our previous study in line 66

2. In the sequencing section please include the sequencing depth and also indicate if the reads were short or long with their size/size range.

Comment: We have updated section 4.6. Genomic Analysis to include “The depth of coverage of the sequencing runs ranged from 70 to 190X, with most exceeding 120X coverage, and the read length per sample is 2 x 50 bp.” In lines 596 -597.

3. Please indicate which is the reference genome chosen.

Comment: The reference genome used for this was indicated and described in 4.1. Media conditions and experimental layout from lines 589 to 591

4. Please explain why only certain metals were used and substantiate the choice of their concentrations with suitable references.

Comments: The choice of heavy metals used for these studies was based on their mechanism of action. As per our previous study, by “Jeje O, Ewunkem AJ, Jeffers-Francis LK, Graves JL. Serving Two Masters: Effect of Escherichia coli Dual Resistance on Antibiotic Susceptibility. Antibiotics 2023; 12 ”, cost, and availability.

5. Fig 1: Escherichia coli should be in italic.

Comments: We have implemented this correction in line 98

6. See Figure 8e). Delete the extra full stop.

Comments: We have implemented this correction in line 273

Round 2

Reviewer 1 Report

Comments and Suggestions for Authors

Ok